# CAR-T after Stem Cell Transplantation in B-Cell Lymphoproliferative Disorders: Are They Really Autologous or Allogenic Cell Therapies?

**DOI:** 10.3390/cancers13184664

**Published:** 2021-09-17

**Authors:** Ariadna Bartoló-Ibars, Mireia Uribe-Herranz, Guillermo Muñoz-Sánchez, Cristina Arnaldos-Pérez, Valentín Ortiz-Maldonado, Álvaro Urbano-Ispizua, Mariona Pascal, Manel Juan

**Affiliations:** 1Immunology Service—CDB, Hospital Clínic de Barcelona, 08036 Barcelona, Spain; bartolo@clinic.cat (A.B.-I.); gumunoz@clinic.cat (G.M.-S.); arnaldos@clinic.cat (C.A.-P.); mpascal@clinic.cat (M.P.); 2Institut d’ Investigacions Biomèdiques August Pi i Sunyer (IDIBAPS), Hospital Clinic de Barcelona, 08036 Barcelona, Spain; uribe@clinic.cat (M.U.-H.); aurbano@clinic.cat (Á.U.-I.); 3Hematology Service—ICMHO, Hospital Clínic de Barcelona, 08036 Barcelona, Spain; vortiz@clinic.cat; 4Faculty of Medicine, Universitat de Barcelona, 08036 Barcelona, Spain

**Keywords:** B-cell malignancies, allo-HSCT, CAR-T, chimerism, allogenic

## Abstract

**Simple Summary:**

B-cell malignancies, like leukemias and lymphomas, are neoplasms that emerge from the malignant proliferation of B cells. Hematopoietic stem cell transplant (HSCT) is an effective medical treatment for these malignancies, but unfortunately, relapse of the disease after HSCT remains a challenge and is associated with poor long-term survival. A cell-based immunotherapy, the chimeric antigen receptor (CAR) T-cells, has proven to improve the clinical outcome of relapsed/refractory HSCT B-cell lymphoproliferative disorders patients in clinical trials. Even though results are promising, in this review, we discuss about the importance to determine T cell chimerism in HSCT patients, the origin of the manufactured CAR T-cells (autologous vs. allogenic) and the future perspective of the CAR-T cells in transplanted patients.

**Abstract:**

Allogenic hematopoietic stem cell transplantation (allo-HSCT) is one of the standard treatments for B-cell lymphoproliferative disorders; however, deep relapses are common after an allo-HSCT, and it is associated with poor prognosis. A successful approach to overcome these relapses is to exploit the body’s own immune system with chimeric antigen receptor (CAR) T-cells. These two approaches are potentially combinatorial for treating R/R B-cell lymphoproliferative disorders. Several clinical trials have described different scenarios in which allo-HSCT and CAR-T are successively combined. Further, for all transplanted patients, assessment of chimerism is important to evaluate the engraftment success. Nonetheless, for those patients who previously received an allo-HSCT there is no monitorization of chimerism before manufacturing CAR T-cells. In this review, we focus on allo-HSCT and CAR-T treatments and the different sources of T-cells for manufacturing CAR T-cells.

## 1. B-Cell Lymphoproliferative Disorders and 1st Line of Treatment

B-cell malignancies are a diverse group of neoplasms that emerge from the malignant proliferation of B cells during their different stages of development [1], and they include lymphomas and leukemias [2]. The first line of treatment includes chemotherapy, but proposals of chemotherapy vary depending on the subtype of disorder and multiple patient factors [3], new chemoimmunotherapy drugs, such as ibrutinib and imatinib, as well as hematopoietic stem cell transplant (HSCT) [4,5,6,7].

HSCT is one of the most effective medical treatments for most hematological malignancies [8], as well as for some non-malignant conditions such as autoimmune disorders [9]. HSCT can be classified into two types of procedures: (a) autologous HSCT (auto-HSCT, when stem cells are collected from the recipient) and (b) allogeneic HSCT (allo-HSCT, when stem cells come from another individual who becomes the donor). In both, hematopoietic stem cells can be collected from peripheral blood (PB), bone marrow (BM), or umbilical cord (UC). Hematopoietic cell transplantation (HCT) in Europe continues to rise with 48,512 HCT in 43,581 patients, comprising 19,798 (41%) allogeneic and 28,714 (59%) autologous, reported by 700 centers in 51 countries during 2019. Main indications were myeloid malignancies (10,518 allo-HSCT; 246 auto-HSCT), lymphoid malignancies (5255 allo-HSCT; 22,640 auto-HSCT), and nonmalignant disorders (2604 allo-HSCT; 569 auto-HSCT) [10]. In this review, we focus on allo-HSCT.

Identifying an HLA-compatible donor for allo-HSCT is an absolute prerequisite to perform this procedure. HLA compatibility with the donor is usually defined by high-resolution typing (four digits at the amino acid level) for ten genes, two genes for each HLA-A, HLA-B, HLA-C, HLA-DR, and HLA-DQ locus [11], one coming from the paternal and the other from the maternal inheritance. HLA-identical sibling donor is the ideal choice when available for allo-HSCT. If that option is not possible, an HLA-compatible unrelated donor is proposed [12]. An increase in donor-recipient HLA disparity in HLA-A, HLA-B, HLA-C, or HLA-DRB1 is associated with poorer outcomes after unrelated donor transplantation [13]. Haploidentical HSCT (with at least one matched familial haplotype) has also been introduced as a successful option.

Given that the aim of allo-HSCT is to remove the patient’s malignant hematopoietic cells, a conditioning regimen based on chemotherapy or radiation is administered. This depletion of the patient’s bone marrow stem cells (and usually also partially tumor cells) induces severe pancytopenia, requiring donor-derived healthy stem cells to re-establish a new blood cell production system. These cells are collected from the donor BM, PB, or CBU and subsequently infused after the conditioning into the receptor’s bloodstream. Once donor stem cells have reached the receptor bone marrow, these cells will reproduce themselves, giving rise to healthy blood cells, including allogenic lymphocytes. During this process of hematopoietic recovery, bacterial, viral, and fungal infections could be favored because of leucopenia, and after recovery of allo-lymphocytes, graft versus host disease (GvHD) can become the main serious life-threatening complication due to the action of these donor’s cells reacting against the recipient’s tissues (mainly gastrointestinal tract, lungs, liver, and skin).

In contrast with the problem of GvHD, allo-HSCT has a much lower relapse risk when it is compared with auto-HSCT, because of the immunological graft-versus-tumor (GvT) effect [14]. GvHD is responsible for the main adverse effect, but its therapeutic effect, GvT, is the result of the eradication of the patient’s remaining malignant cells by immunocompetent T-cells of the donor. In this line, current work is focused on enhancing the GvT effect without increasing the risk of GvHD.

Although allo-HSCT is considered the cornerstone in the treatment of hematological malignancies, relapse of the hematological disease after allo-HSCT remains a challenge and is associated with poor long-term survival [15], highlighting the need for new approaches in treating relapsed/refractory (R/R) B-cell malignancies patients.

## 2. CAR-Redirected T-Cells (CAR-T) Offer a New and Promising Cell-Based Immunotherapy

Immunotherapy, the most recent and effective cancer therapy, relies on using our own immune system to recognize and eliminate cancer cells. Beyond SCT, there are several types of additional immunotherapies: monoclonal antibodies, cancer vaccines, or the most successful ones so far, immune checkpoint inhibitors [16] and T-cell transfer therapy [17], including adoptive cell therapy using tumor-infiltrating lymphocytes (TILs) [18] or chimeric antigen receptor (CAR) T-cells [19].

Adoptive cell therapy depends on the capacity of the T-cell to fight and destroy cancer cells. Using gene transfer technologies (transfection of a gene inside cells through vectors or gene editing), it has been possible to genetically modify T-cells to stably express defective or “new” molecules. Although replacement of the wild-type gene in mutated T-cell precursors was the initial gene transfer proposal to treat some severe immunodeficiencies [20], the most revolutionary changes that gene therapy is providing on immune cells arrive from the modification of the T-cell surface by expression of antibody-derived chimeric molecules with binding domains that confer novel antigen recognition in a major histocompatibility complex (MHC)-independent manner. CARs are the most frequent application of these approaches. The extracellular domain of the CAR derives from a monoclonal antibody directed against the tumoral target. This extracellular domain consists of a single-chain variable fragment (scFv), which is the result of combining the variable heavy (VH) and the variable light (VL) chain by a linker. The scFv domain is anchored to the transmembrane region/domain by an aminoacidic hinge or spacer (Figure 1).

The activation of CAR-T cells is the result of the recognition of the antigen by the scFv region, which concludes in the clustering and immobilization of the CAR molecules [21]. Intracellularly, the co-stimulatory domain chosen (typically sequences of CD28 or 4-1BB co-stimulatory regions) will determine the intracellular signaling and also could influence CAR dynamics, with the presence of a 4-1BB co-stimulation domain conferring slower expansion and longer persistence compared with the presence of a CD28 co-stimulation domain, which leads to rapid expansion but less durability [22,23]. Increased proliferation and expansion have also been associated with the 4-1BB co-stimulation domain when compared with CD28 [24]. Additionally, other co-stimulatory domains such as OX40, CD27, and inducible T-cell co-stimulator (ICOS) have been tested [21].

Subsequently, tyrosine-based activation motifs (ITAM) domains on the CD3ζ chain are phosphorylated, fostering the intracellular signaling through ZAP70 protein [25], which promotes CAR proliferation, cytokine release, metabolic alterations, and cytotoxicity. The release of granules with perforin and granzyme is the main mechanism associated with the CAR antitumoral effect. However, other death receptors such as BH3-interacting domain death agonist (BID) and FAS-associated death domain protein (FADD) have also been associated with CAR antitumoral function [25,26,27].

The process to manufacture CAR-T can take several weeks and can be performed in different ways. In our experience with CAR-T production from an academic clinical trial with a locally developed anti-CD19 CAR-T (ARI-0001) [28], patients provide the starting material: their own lymphoapheresis products; subsequently, in our proposal, this apheresis product is subjected to CD4- and CD8-positive selection, cultured and activated using anti-CD3 and anti-CD28 antibodies, by using a bioreactor that manages all the process under a close system by a sterile tubing set. After 24 h of activation, T-cells are transduced using our CAR-containing lentivirus and maintained in culture for expansion with IL-7 and IL-15 along 9–12 days.

Anti-CD19 CAR T-cells have shown outstanding results in clinical trials against leukemia and lymphoma [22,23,29,30,31,32], inducing high rates of response in patients with relapsed/refractory (R/R) B-cell lymphoproliferative disorders. In B-ALL, CD19-CAR therapy has achieved 60–93% complete responses (CRs) across several studies (reviewed in [33]) (as explained in more depth below). In R/R non-Hodgkin lymphoma (NHL) patients, CD19-CAR therapy achieved 40–90% of CR in heavily pretreated patients as reported by some of the main clinical assay JULIET [34], ZUMA-1 [35], and TRANSCEND NHL 001 [36].

This led to the approval by the US Food and Drug Administration (FDA) and the European Medicines Agency (EMA) of two CAR-T cell therapies, Tisagenlecleucel (Kymriah) for relapsed/refractory (R/R) B-cell acute lymphoblastic leukemia (ALL) and R/R diffuse large B-cell lymphoma (DLBCL), and Axicabtagene Ciloleucel (Yescarta) for R/R DLBCL, primary mediastinal large B-cell lymphoma, high-grade B-cell lymphoma, and DLBCL arising from follicular lymphoma. Recently, brexucabtagene autoleucel (Tecartus) for R/R mantle cell lymphoma and lisocabtagene maraleucel (BREYANZI) for DLBCL, high-grade B-cell lymphoma, primary mediastinal large B-cell lymphoma, and follicular lymphoma grade 3B were approved by FDA as well. At a more local level, there is also our CART19 cell therapy, a fully academic CAR approved by the Spanish Agency of Medicines (AEMPS) (ARI-0001) for R/R ALL, approved under the European “Hospital Exemption” rule.

Cell-based immunotherapies have revolutionized the approach to treat cancer. One of the main advantages of this therapy is that CAR T-cells bind their tumor target antigen specifically and efficiently enough to eliminate cancer cells, in an HLA-independent manner [37]. This prevents the tumor escape mechanism of downregulating the expression of HLA to avoid T-cell immune surveillance. Unlike chemotherapy or radiotherapy, the specificity of this approach avoids the unnecessary killing of healthy cells and tissues. Furthermore, the CD19 target of the CAR19 covers most of the B-cell malignancies, over 95%, making it extremely versatile and useful. Finally, a big difference with conventional cancer treatments such as chemotherapy and radiation is that CAR-T is considered living drugs, capable of proliferating and remains for years in the patient [38]. Despite the outstanding results of the CAR19 therapy, there are some limitations that currently are impairing its general use. One of the most important is the availability and fitness of T-cells used to manufacture the CAR19, as patients heavily treated might suffer from T-cell lymphopenia, and that can prevent them from being eligible for this therapy. This issue could potentially be solved using modified allogeneic T-cells to avoid GVHD manufacturing the CAR-T. Another issue related to the patient’s condition is the time that currently is necessary to generate the CAR-T; some rapidly progressive patients might not be able to wait. In some cases, clinicians can perform bridge chemotherapy to lower the tumor burden temporarily.

Another disadvantage is the CAR-T cell-associated toxicities such as cytokine release syndrome (CRS) and the immune effector cell-associated neurotoxicity syndrome (ICANS), both clinically manageable. Finally, another big hurdle is the tumor escape by target antigen loss, relapse patients with CD19 negative disease have been described in several clinical trials [39]. Strategies to overcome this problem include targeting multiple antigens with a bispecific or a tandem CAR [40,41].

Although clinical trials testing CAR-T targeting CD22 or BCMA are also presenting encouraging results for B-ALL [42] and multiple myeloma [43], and additional information will also arrive soon from the recently FDA-approved CAR-BCMA [44,45], in this review, we focus on the already approved anti-CD19+ CAR T-cell therapies.

## 3. Combination of HSCT and CAR-T Treatments

HSCT and CAR-T therapy are two different but frequently combinatorial approaches in the treatment of patients with R/R B-cell lymphoproliferative disorders: (a) the first approach considers CAR-T the firmest option for treatment after an unfruitful SCT, refractoriness to HSCT is one of the main criteria for the indication of approved commercial CARTs; (b) on the other hand, because relapses are common after an HSCT in patients with B-cell lymphoproliferative disorders with minimal residual disease (MRD+), an alternative strategy to overcome these relapses is to treat patients with CAR-T before transplantation, as a bridge towards allo-HSCT.

Returning and delving into the first combinatorial situation (a), the use of CAR-Ts directed against CD19, which is expressed in over 95% of B-cell malignancies, is a clear indication obtaining great results in clinical trials in post-transplanted relapsed patients, where other non-B-cell specific therapies options, such as salvage chemotherapy, have very reduced effectiveness. Although CAR19 on-target off-tumor effect exists, the B-cell aplasia and hypogammaglobulinemia, it is proven that patient can receive intravenous administration of immunoglobulin to prevent infection and lead a normal life. CAR-Ts directed against CD19 lyse tumor cells via direct T-cell/tumor-cell interactions, thanks to the specific tumor antigen recognition, and produce cytokines such as perforin or granzyme that increase their antitumoral capabilities [21]; therefore, tisagenlecleucel (Kymriah, CTL019) was approved for the treatment of pediatric and young patients with B-ALL in relapse post-allo-HSCT or in later relapse. The same product was approved for adults with DLBCL who failed two or more lines of standard treatment, including HSCT. The clinical trials reported CR rates up to 85% and 65%, respectively [46,47]. Axicabtagene ciloleucel (Yescarta, KTE-C19) was approved as a treatment for the same DLBCL indication as CTL019. Clinical data reported CR rates up to 65% in children [48] (Table 1). Both products improved the overall survival (OS) in those patients who failed the standard treatments such as allo-HSCT [49].

Other clinical trials such as ARI-0001 [32], included patients with CD19+ R/R B-cell malignancies (adult and pediatric B-ALL, CLL and NHL), of which 35 (74.5%)—29 adults (80.6%) and 6 pediatrics (54.5%)—received an haploidentical allo-HSCT (10/10); obviously, the final infused product shows the same HLA-genotyping than those of the donor. This phase 1 clinical trial reported a complete response rate (CRR) up to 79% in children and 85.2% in adults, respectively. Gardner and colleagues [50], included CD19+ ALL young and children’s patients, of which 62% at least received one previous allo-HSCT. This phase 1 clinical trial revealed that 29 out of 45 patients (64.4%), in which CAR T-cells persist, did not need a consolidative allo-HSCT in contrast as others studies reported: Curran et al. [51] revealed that 83% of young patients included in the clinical trial underwent allo-HSCT as a consolidative treatment (in this study, the other patients who did not receive a consolidative allo-HSCT were because of organ dysfunction and MRD+) (Table 1). For the second combinatorial option (b), Davila et al. [52] used CAR T-cells as a bridge to allo-HSCT even if 25% of patients received a previous allo-HSCT before CAR therapy (allo + CAR-T + allo double combinatorial proposal); no relapse after this second allo-HSCT was reported in 44% of these patients. Contrary to these positive data that indicate the benefit of CAR-T therapy as a bridge to allo-HSCT, Park et al. [31] did not observe improvement in those patients who were treated with CAR T-cells and allo-HSCT (Table 1). So, the results of this approach are controversial and there is no defined protocol to determine which transplanted patients need a consolidative allo-HSCT post-CAR-T therapy (allo + CAR-T + allo).

## 4. Chimerism before Manufacturing CAR-T

As stated before, allo-HSCT has become an essential treatment for B-cell lymphoproliferative disorders. Even though this therapy improves the clinical outcomes in combination with CAR-T therapy, surviving recipients’ cancer stem cells could facilitate the re-emergence of a malignant cell clone, and consequentially increase the relapsing disease risk [53,54,55].

For this reason, it is important to monitor donor-recipient chimerism and determine the origin of HSC in recipients to predict the outcome of the allo-HSCT [56]. Nonetheless, it has been reported that chimerism could be mixed (<95%) or persists even though the patient relapses (<95%) [57]. It has been demonstrated that chimerism is a dynamic process, as patients with complete chimerism post-transplant can later develop a “mixed chimerism” and conversely [54]. Hence, it is important to monitor chimerism in patients post-transplant. A standardized methodology was developed to analyze chimerism from peripheral blood samples and is based on 13 short tandem repeats (STR) by polymerase chain reaction (PCR) [58,59]. In this setting, the manufacturing of CART19 cells from a patient could be autologous, allogeneic, or pseudo-allogeneic, depending on the % of chimerism.

Siglin et al. [60] described the concept of “pseudo-allogeneic” CAR-T as modified T-cells collected from an allo-HSCT recipient who displays ≥ 95% of chimerism. A single-center study using pseudo-allogenic CAR-T in six patients with R/R B-cell lymphoma apparently seemed to be safe and well-tolerated. Moreover, in a study by Qing and colleagues, CAR-T functionality was assessed in vitro in nine relapsed B-ALL patients after an allo-HSCT. These patients, which present a low percentage of chimerism, restored chimerism after 12 days of T-cell culture. CAR T-cells showed strong cytotoxicity against CD19+ target cells, NALM6 [61]. Further experiments are needed to prove the efficacy and safety of these pseudo-allogeneic CAR-T.

## 5. Autologous or Allogenic CAR-T

Currently, there is an active area of investigation regarding the importance of the origin of the manufactured CAR T-cells. One approach is to obtain T-cells from the recipient’s allo-HSCT donor (allogenic) and the other one from the patient itself (autologous) (Figure 2).

One of the requirements of using CAR-T derived from the HSC’s donor is the complete HSC engraftment before leukapheresis. This clearly begs the question of whether this CAR-T product should be considered allogeneic or autologous. Complete engraftment implies that the receptor’s hematopoietic system should be considered equal to donor’s, so one of the main issues of allogeneic CARTs such as alloimmunization (rejection of allo-CAR-T by receptor’s immune system) could be avoided, providing a major benefit from this procedure. Currently, CAR T-cells are considered autologous, independently whether the patient has received or not an allo-HSCT and the level of chimerism of the T-cells in the transplanted patients; however, from the biological point of view, CAR-T could only be considered autologous if the T-cells are from a patient that did not receive a previous allo-HSCT. For all the other possible scenarios, we should be talking about allogenic CART cells, where T-cells can be sourced from the same donor as the allo-HSCT, or directly from the allo-HSCT transplanted patient. The origin of T-cells used to manufacture the CAR-T has an outstanding impact on the overall process, from differences in the biological activity to potentially different costs in production. For instance, T-cells obtained from a heavily treated patient might be limited in numbers, and that can affect the administered dose, as well as the possibility to reinfuse CAR T-cells in case of necessity. Moreover, the manufacturing process is longer than using T-cells from a healthy donor, and more prone to failure. This represents an important issue for patients that show disease progression before CAR T-cells are available [62]. On the other hand, T-cells from healthy donors, even though not limited in numbers, could cause GVHD or be rapidly eliminated by the receptor’s immune system [62,63]. In any case, the use of T-cells in patients after allo-HSCT for manufacturing CAR-T opens an additional aspect of controversy. As commented, when T-cells are obtained and manufactured from the transplanted patients who will receive the infusion, these are considered autologous, even when these cells are genetically identical to the donor of allo-HSCT. In fact, in this situation, it is often possible to have access to the donor to obtain healthy T-cells. Unexpectedly, if the CAR-T development is made from newly obtained cells from the donor, this product will be considered by drug agencies an allogenic advanced therapy medicinal product (ATMP), different from the autologous ATMP developed from the transplanted patients. This aspect is very relevant because the current regulatory interpretation prevents the use of “better” cells from healthy donors who never received treatment previously and could provide functionally better CAR T-cells for the patients (or at least with lower accumulative cell potential injuries from previous treatments received by the patients).

Several clinical trials have compared CAR T-cells sourced from patients who received or did not receive an allo-HSCT. These studies achieved a similar minimal-residual disease and complete remission between groups [31,50]. In addition, there were no differences in treatment-related adverse events such as cytokine release syndrome and neurotoxicity. Although these clinical trials reported similar clinical outcomes, it is less common to use T-cells sourced from a patient who did not receive an allo-HSCT. As stated, allo-HSCT is one of the first standard treatments for B-cell malignancies, but we should consider that not all patients have the option of an HLA-compatible stem cell donor, or this procedure is not recommended for their clinical condition.

Numerous studies have reported that relapsed allo-HSCT patients with B cell malignancies could benefit from post-HSCT CAR-T infusion from the original HSC donor. Brudno et al. [64] conducted a clinical trial where CAR T-cells, targeting CD19, were obtained from each recipient’s allo-HSCT donor. This could lead to a major advantage as these cells have not been subjected to previous therapies administered to the patient to diminish the tumoral burden, but that could also diminish its numbers or function. According to the authors, no chemotherapy or other therapies were administered due to concern that the introduction of CAR T-cells into a recipient with depleted lymphocytes might cause severe GVHD. Eight of twenty treated patients obtained remission, which included six complete remissions and two partial remissions. The response rate was highest for acute lymphoblastic leukemia, with four of five patients obtaining minimal residual disease-negative complete remission. Responses also occurred in chronic lymphocytic leukemia and lymphoma. It is especially relevant to the fact that no acute GVHD was reported, suggesting that this strategy could reinforce the GVT effect without increasing the risk of GVHD. These results are in line with previous studies that reported regression of B-cell malignancies resistant to standard donor lymphocyte infusions without causing GVHD [65,66].

Focusing on the T-cells sourced from transplanted patient, there are still some challenges for manufacturing CARTs because the biological characteristics of these T-cells could negatively impact by the previous lines of treatment [62]. Moreover, T-cells could be dysfunctionally associated with immunosuppression treatment derived from transplants. Further, there are no data about chimerism before manufacturing CAR-T from a transplanted patient. Considering the percentage of chimerism could be important for having an idea about the T-cell fitness and indirectly associate it with a better or worse outcome with the manufacturing CARTs (transduction and expansion of CAR T-cells).

As it has been mentioned before, pseudo-allogenic CAR-T, from patients with 95% of chimerism, is well-tolerated and safe [60]. Although these CARTs have more risk of being alloreactive and increase toxicity, this study did not observe that. Even though they obtained good clinical outcomes in 6 patients, there is a need for further clinical trials using pseudo-allogenic CAR-T. Although there is a monitoring of the chimerism to assess the engraftment of allo-HSCT patients, there is a lack of data about chimerism before manufacturing CAR-T. The analysis of the T-cell chimerism after leukapheresis from a transplanted patient may be useful for predicting the quality and quantity of CAR-T product, adding a relevant layer of information to foresee the CAR-T therapy outcome. Currently, most clinical trials using CAR T-cells as a therapeutic approach use allo-HSCT patients T-cells, between 20–80% of the patients depending on the study (Table 1). As results seem to be related mainly to the “functional quality” of the T-cells, regulators should probably introduce changes in their interpretation to help in the use of the best product for each patient.

In summary, a chimerism analysis from the leukapheresis product would give us the ability to better compare clinical results between CAR-T generated from allo-HSC transplanted patients and CAR T-cells produced from matching HSCT-donor, trying to better define in an easy way the best product for each patient. “Starting material” for any product should be mainly defined by genetic characteristics, ahead of the extraction site (donor o patient) considerations. The final success of CAR-T therapy is obviously dependent on this origin, above general normative considerations.

## 6. Future Perspective

Beyond the good efficacy of the CAR-T therapy, the safety of CAR-T after allo-HSCT is one of the main concerns of the field; as CAR-T therapy is at the beginning of its development, there is limited follow-up and long-term safety and efficacy data. Even though most of the patients fully recover, life-threatening complications such as CRS and ICANS can send patients to intensive care units, and a small percentage of them die from these complications. Moreover, the infusion of allogeneic CAR-T cells shows a GVHD incidence of 10%, and it is still not clear which factor is the leading cause; it could be the source of T-cells or CAR-T cell population, among other options. It is also worth highlighting clinical trials published included a very restricted selection of patient population, representing a favorable selection, and hence a positive bias. Even considering all those unknowns and concerns, the truth is that CAR-T has revolutionized cancer treatment. A key point for the near future is to standardize the CAR-T product to expand its application. Many clinical trials have used T-cells sourced from the patient, whether previously allo-HSCT or not. Properties such as number of T-cells, viability, T-cell phenotype, CAR expression, and cytotoxic effects are affected and so will be different for each product [62,67]. The possibility of a universal CAR-T system, which would not require HLA matching and could be ready to use, off-the-shelf, would be key to reach the standardization and is the next big challenge of the CAR-T in lymphoproliferative disorders [62]. The off-the-shelf CAR-T would allow new CAR-T applications, expanding from the range of cancers that will be treatable to the patients that will be eligible for treatment. Importantly, off-the-shelf therapies can cut down the manufacturing time, bring down the current high price, and increase the accessibility and fitness of the starting material, namely the T-cells [62,67]. The last one is especially important, as T-cell fitness has revealed itself as a key factor for the efficacy and safety of CAR-T [68]; however, two important issues need to be addressed: the first one is that the immune system might recognize the off-the-shelf CAR-T as foreign and eliminate them rapidly, so persistence might be a concern; the second one is the development of GvHD. To avoid these major problems, there are some strategies that are being developed, such as the use of T-cells from the allo-HSCT donor, gene-editing molecules that are key in allo-recognition such as β2-microglobulin, or the alpha-beta TCR, but there is also the possibility to use a non-αβ cell, such as NK or γδ T-cells.

Finally, immunotherapy treatments are positioned as the cornerstone of B lymphoproliferative disorders treatments, allo-HSCT, CAR-T therapy, and combinations, and are the best options for the future. Even though the most common application is CAR-T after the failure of an allo-HSCT, some studies are already testing CAR-T before transplantation as a bridge towards allo-HSCT. This strategy might allow an effective way to induce remission, reducing tumor burden, and therefore improving the outcome of the subsequent allo-HSCT [49,69,70]; however, CAR-T therapy is only at the beginning of its application, and many problems and difficulties are still avoiding its widespread application; standardization of the T-cell source to manufacture CAR-T is an active area of research that in the next few years will undoubtedly benefit the efficacy of CAR-T therapy.

## 7. Conclusions

The origin of the “autologous” cells used for CAR-T treatment is a main aspect to have in mind when this treatment is used, specially when the patient arrives to this advance therapy after an allo-HSCT. Many CAR-T therapy clinical trials have used T-cells sourced from the patient, whether previously allo-HSCT or not, hence T cells properties are different and further studies are needed to understand the importance and consequences of the T cell source and its chimerism in this setting. Finally, standardization of the CAR-T product could consolidate the success of the therapy and to expand its application.

## Figures and Tables

**Figure 1 cancers-13-04664-f001:**
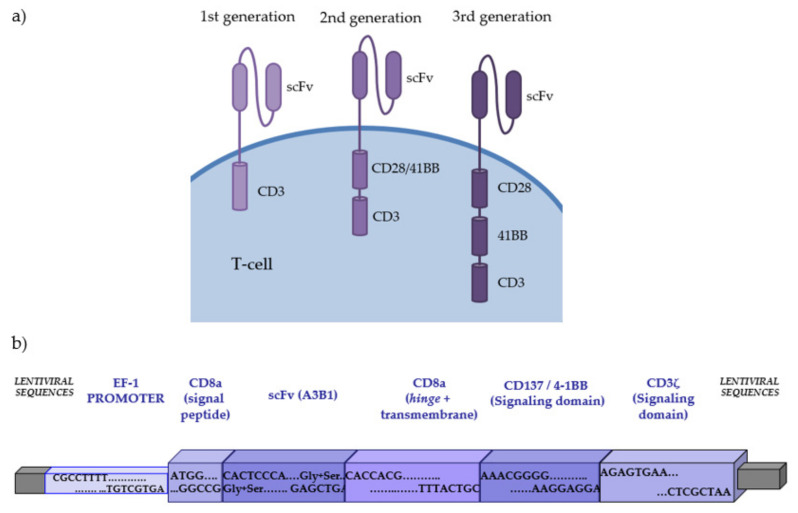
(**a**) Different generations of CAR-T cells. First-generation CAR-T cells include an intracellular domain. Second-generation CAR-T cells incorporate an additional co-stimulatory domain. Third-generation CAR-T cells include multiple co-stimulatory domains. (**b**) Structure of second-generation anti-CD19 CAR developed at the Immunology Department of Hospital Clínic de Barcelona.

**Figure 2 cancers-13-04664-f002:**
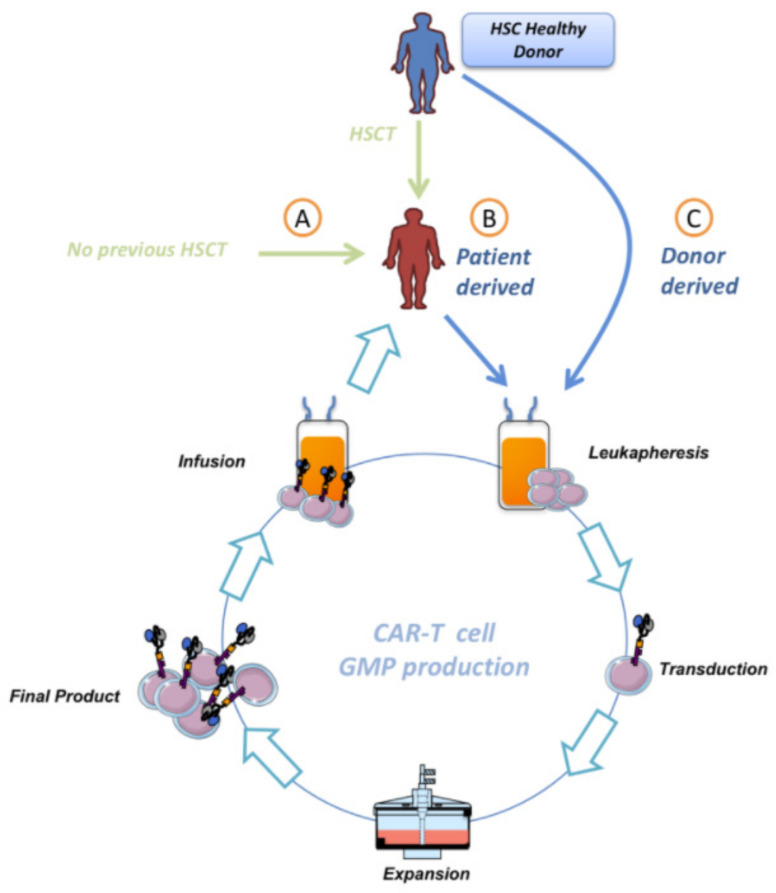
Different protocols in CAR-T therapy for B-cell malignancies. T-cells from patients with B-cell malignancies could be obtained from patients after HSCT (B) or without previous HSCT (A). Further, T-cells could be obtained from each recipient’s allo-HSCT healthy donor (C).

**Table 1 cancers-13-04664-t001:** Summary of results by relevant clinical trials with CAR-T anti-CD19+.

Patients	Product	scFv	scFv Origin	Coestimulatory Domain	CRR, % (CI 95%)	PFS/EFS, Median (CI 95%)	OS, Median (CI 95%)	Previous allo-HSCT, %	Post-allo-HSCT in CR, %	Reference
**PEDIATRICS +/− YOUNG ADULTS**										
11 kids (up to 25y)	ARI–0001	A3B1	Murine	4–1BB	79 (54–94)	18.1 mo (14.5–ND) 82% (59–100) at 1y	NA (7.1–NA) 78% (50–100) at 1y	55	NA	[32]
75 kids (up to 25y)	CTL019	FM63	Murine	4–1BB	81 (71–89)	50% (35–64) at 1y	76% (63–86) at 1y	61	14	[23]
53 kids	KTE–C19	FM63	Murine	CD28	61	49% at 18 mo	52% at 10 mo	35	75	[48]
45 kids (up to 25y)	JCAR017	FM63	Murine	4–1BB	93	51% (37–70) at 1y	69.5% (56–87) at 1y	62	28	[50]
25 kids (1–23y)	19–28z	SJ25C1	Murine	CD28	75	NA	NA	20	83	[51]
**ADULTS**										
27 adults (>18y)	ARI–0001	A3B1	Murine	4–1BB	85.2 (66–96)	9.4 mo (3.3–20.2) 34% (12–57) at 1y	20.2 mo (12.8–NA) 65% (40–89) at 1y	81	NA	[32]
53 adults	19E3/1928z	19E3 ab	Murine	CD28	83 (70–92)	50 (at 6 mo)	12.9 mo (8.7–23.4)	36	39	[31]
35 adults	CTL019	FM63	Murine	4–1BB	69 (51–83)	5.6 mo	19.1 mo (6.2–NA)	37	38 (9 out of 24)	[47]
16 adults	19–28z	SJ25C1	Murine	CD28	88	NA	NA	25	44	[52]

CRR, complete response rate; PFS/EFS, progression-free survival/event-free survival; OS, overall survival; y, years; mo, months; NA, not available.

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
