# Peer review of "CAR-T after Stem Cell Transplantation in B-Cell Lymphoproliferative Disorders: Are They Really Autologous or Allogenic Cell Therapies?"

_cancers, 2021, doi:10.3390/cancers13184664_

Round 1

Reviewer 1 Report

The issues of this reviewer have been adequatley addressed.

This manuscript is a resubmission of an earlier submission. The following is a list of the peer review reports and author responses from that submission.

Round 1

Reviewer 1 Report

I am satisfied to see the improvement made in the revised version of this manuscript. 

Reviewer 2 Report

This is a timely review summarizing concepts arising from the availability of CAR-T treatment in relapsed B-cell malignancies. A particular focus is given to the sequence of HSCT and subsequent CAR-T and vice versa. The subject is timely, relevant and bears unanswered questions which are addressed in this review. The review is easy to read, well structured, and it contains no obvious misinformation detectable to this reviewer.

- For the reader, one is most tempted to learn the results of the HLA typing of the presented Spanish study which may answer most of the questions (are CAR-T cells produced after allo transplant autologous, mixed, or allogeneic ?). It is kind of frustrating to produce a review with all questions asked, then to present a completed study possibly asking all these questions - but to hold back the results, most likely for a different publication. The authors are invited to give an insight in these results.